# “Repurposing” Disulfiram in the Treatment of Lyme Disease and Babesiosis: Retrospective Review of First 3 Years’ Experience in One Medical Practice

**DOI:** 10.3390/antibiotics9120868

**Published:** 2020-12-04

**Authors:** Jiachen Gao, Zhaodi Gong, Dawn Montesano, Erica Glazer, Kenneth Liegner

**Affiliations:** 1College of Arts and Sciences, Boston University, 725 Commonwealth Avenue, Boston, MA 02215, USA; gaojiachen10@gmail.com; 2CT Integrated Pain Consultants, LLC, 60 Katona Drive, Suite 25, Fairfield, CT 06824, USA; abatapain@gmail.com; 3P.C. Lyme Borreliosis & Related Disorders, 592 Route 22, Suite 1B, Pawling, NY 12564, USA; dawnmontesano83@gmail.com (D.M.); eglazer5@optimum.net (E.G.); 4Northwell System, Northern Westchester Hospital Center, Mount Kisco, NY 10549, USA; 5Nuvance Health System, Sharon Hospital, Sharon, CT 06069, USA

**Keywords:** disulfiram, chronic Lyme disease, post-treatment Lyme disease syndrome (PTLDS), persistent infection, antibiotic treatment failure, relapsing infection, babesiosis, bartonellosis

## Abstract

A total of 71 patients with Lyme disease were identified for analysis in whom treatment with disulfiram was initiated between 15 March 2017 and 15 March 2020. Four patients were lost to follow-up, leaving 67 evaluable patients. Our retrospective review found patients to fall into a “high-dose” group (≥4 mg/kg/day) and a “low-dose” group (<4 mg/kg/day). In total, 62 of 67 (92.5%) patients treated with disulfiram were able to endorse a net benefit of the treatment with regard to their symptoms. Moreover, 12 of 33 (36.4%) patients who completed one or two courses of “high-dose” therapy enjoyed an “enduring remission”, defined as remaining clinically well for ≥6 months without further anti-infective treatment. The most common adverse reactions from disulfiram treatment in the high-dose group were fatigue (66.7%), psychiatric symptoms (48.5%), peripheral neuropathy (27.3%), and mild to moderate elevation of liver enzymes (15.2%). We observed that although patients on high dose experienced a higher risk for adverse reactions than those on a low dose, high-dose patients were significantly more likely to achieve enduring remission.

## 1. Introduction

Lyme disease is caused by an infection of *Borrelia burgdorferi* bacteria transmitted through tick bites. Symptoms commonly include fever, fatigue, headache, and a rash. If left untreated, serious consequences affecting musculoskeletal, cardiac, neurologic, and other systems may arise. Lyme disease is the fastest-growing vector-borne disease and is an issue of great public concern. The Center for Disease Control and Prevention (CDC) estimates approximately 300,000 annual cases of Lyme disease in the United States [1]. The recent $25 million collaboration between the U.S. Department of Health and Human Services (HHS) and the Steven and Alexandria Cohen Foundation: “LymeX”, the largest Lyme private–public partnership in history, speaks to the extensive interest of patients and practitioners seeking additional options for the treatment of Lyme disease above and beyond antibiotics alone [2].

Lyme disease has been traditionally treated with short courses of antibiotics. Although antibiotic therapy has been helpful in resolving the infection in many individuals and yielding improved quality of life for many patients, it became apparent to us and many other healthcare providers that antibiotics frequently were suppressive and that many patients relapsed when antibiotics were withheld [3,4,5,6]. When treatment is unsuccessful, the disease is often referred to as chronic Lyme disease. Recently Shor et al. [7] defined chronic Lyme disease as an illness, either treated or untreated, enduring 6 months or longer and due to persistence of *B. burgdorferi sensu lato*. Stricker and Fesler [8] have defined chronic Lyme disease to also subsume non-Lyme borrelioses such as tick-borne relapsing fever agent *Borrelia miyamotoi* as well as other tick- and vector-borne co-infections. Post-treatment Lyme disease syndrome (PTLDS), most precisely defined by Rebman and Aucott [9], consists of a more restricted subset of the “universe” of chronic Lyme disease that lends itself well to research studies. Antibiotic treatments may come with serious and irreversible adverse effects, including potentially fatal line sepsis or *C. difficile* colitis among other complications [10,11]. Additionally, the high cost of intravenous antibiotic therapies is one of the barriers to treatment for many patients. Even after rigorous antibiotic treatment, symptomatic relapse may occur, and abundant peer-reviewed literature demonstrates that this can be due to persistent borrelial infection in some patients [3,4,5,6]. Some researchers believe drug-tolerant persisters to be the cause of these symptoms, among other proposed mechanisms [12,13,14,15,16].

A combination of a rapidly increasing patient population and a small number of treatment options with limited efficacy led researchers to seek to identify agents with activity against *B. burgdorferi* superior to conventionally used antibiotics. Pothineni et al. [17] evaluated a variety of agents in the pharmacopeia by means of in vitro high-throughput screening that might have superior activity against *B. burgdorferi* than commonly used antibiotics such as doxycycline. A high throughput screening of 4366 chemical compounds identified 150 unique compounds that were able to inhibit >90% of *B. burgdorferi* growth at a concentration of <25 μM, 101 of which were approved by the United States Food and Drug Administration (FDA), and 20 of which were ultimately selected for further investigation for safety and potency. Among the 20 molecules with higher bacterial inhibition (>95.25%) than the commonly prescribed doxycycline (94.14%), tetraethylthiuram disulfide (disulfiram) was shown to be the most effective molecule, inhibiting 99.80% of bacteria. Other molecules with comparable bacterial inhibition and minimum concentrations include doxorubicin hydrochloride and epirubicin hydrochloride. However, the latter cancer chemotherapy agents appear to be unsuitable to apply in the treatment of non-malignant conditions.

The findings from this study prompted the off-label use of disulfiram in the first three Lyme disease patients relapsing to traditional treatments with antibiotics [18] (Cases 1, 2 and 3 in that article correspond to Cases 18, 49 and 10, respectively, in the current report). The dramatic improvement we observed in these individuals encouraged us to offer disulfiram as a treatment option to other suitable patients. We identified 67 evaluable patients placed on disulfiram between March 2017 and March 2020 for further analysis, reported here.

## 2. Results

There were 35 females and 32 males, ranging from 17 to 83 years old. In total, 13 patients (19.4%) received more than one course of treatment with disulfiram. Furthermore, 27 patients (40.3%) were receiving ongoing disulfiram treatment at the time of submission of the manuscript for this article (July 2020). Each patient was treated for a range of 6 weeks to greater than 16 months. Retrospective analysis of the 67 patients allowed us to divide them into two categories depending on the average daily disulfiram dosage: “high dose” (i.e., ≥4 mg/kg/day) (Table 1) and “low dose” (i.e., <4 mg/kg/day) (Table 2). Among the 67 patients, 62 (92.5%) endorsed net benefit from the received courses of disulfiram. Totally, 12 of 33 (36.4%) patients who completed one or two courses of high-dose treatment of at least 6 weeks enjoyed enduring remission. As of the time our manuscript was being readied for submission (July 2020), patient 18 has remained clinically well for greater than 3 years and patients 1, 3, 7, 8, 11, 12, 16, and 21 for greater than one year following their last course of disulfiram.

Of 33 high-dose patients, 22 (66.7%) reported fatigue, 16 (48.5%) reported psychiatric symptoms, 9 (27.3%) reported symptoms consistent with peripheral neuropathy, and 5 (15.2%) reported elevated liver enzymes. Liver enzyme elevation required the cessation of treatment in 2 patients but fully resolved in all cases (Table 1). Comparatively, low-dose patients exhibited lower percentages of adverse reactions, 32.4%, 2.9%, 14.7%, and 14.7%, for fatigue, psychiatric symptoms, peripheral neuropathy, and elevated liver enzymes, respectively (Table 2). Fatigue was often described as tiredness, lethargy, lack of energy, exhaustion, or drowsiness. Psychiatric symptoms often presented as emotional instability ranging from hypomania to anxiety and/or depression. In several cases, psychiatric symptoms required cessation of therapy and psychiatric consultation, with symptoms resolving within days to weeks of discontinuing disulfiram. Symptoms consistent with peripheral neuropathy were described as abnormal skin sensations such as numbness, tingling, itchiness, burning, or pain. In nearly all cases, disulfiram-induced peripheral neuropathies resolved fully over weeks to months following cessation of the drug. In one case, mild residuum has remained and not fully resolved.

The difference between the high-dose and low-dose patients in the incidence of adverse reactions as well as achieving enduring remission reached statistical significance (*p* < 0.05) using the χ^2^ test (Table 3). Of note, low-dose patients reported fewer adverse reactions to disulfiram, with the exception of elevated LFT. At this point, we are unable to determine whether the low-dose approach can “enable” an enduring remission because few low-dose patients who fell within our inclusion group had discontinued their course of disulfiram treatment as our manuscript was being submitted. In total, 23 of 27 (85.2%) patients receiving ongoing treatment continued to stay on disulfiram, even at a very low doses ranging from less than 1, up to 2 mg/kg/day to control symptoms of Lyme disease with few and/or minor adverse reactions (Table 1 and Table 2).

Patients 10, 23, 39, 49, and 50 had distinct treatment experiences in that they took both high and low dosages throughout their treatment (Table 1). However, they were included in the high-dose group because their maximum doses were equal to or greater than 4 mg/kg/day. These 5 patients provided valuable information in the development of dosing strategies for future patients. For example, 4 of these patients started with high doses of disulfiram, but due to intolerable adverse reactions, disulfiram treatment was temporarily suspended and reinstituted at lower dosages. Low doses were used for some neurologically or psychiatrically fragile patients, which is why their doses were titrated up so slowly. In some cases, aggressive dose increases actually resulted in destabilization of patients’ conditions, suggesting that they would experience more net benefit by continuing with their lower dose treatment. Other patients had baseline peripheral neuropathies from their underlying disease (e.g., Lyme neuropathy) and were afraid that more aggressive dosing might worsen their neuropathies, hence requesting very cautious, low dosing. Additionally, it became clear that disulfiram provoked in some individuals what was deemed to be profound Jarisch–Herxheimer-like reactions, once again requiring a measured and cautious approach.

Co-infection with babesiosis was present in 15 of 67 patients (*Babesia microti* (*n* = 4); *Babesia duncani* (*n* = 11)) in addition to Lyme disease (Table 4). All of the patients reported improvement after at least one course of disulfiram treatment, including five patients (33.3%) who achieved enduring remission. One patient with motor neuron disease experienced diminished constitutional symptoms with disulfiram but succumbed to his neurologic illness, nonetheless. Another patient, who did not benefit from high doses of disulfiram (7.5 and 3.8 mg/kg/day), did report clinical improvement at a lower dose (0.6 mg/kg/day). Of note, these 15 patients did not take any antimicrobial medications besides disulfiram. It appears that patients who had concurrent Lyme disease and babesiosis showed clinical improvement with disulfiram, alone.

Besides Lyme disease, 12 of 67 (17.9%) patients tested positive for exposure to *Bartonella henselae*. Table 5 illustrates the effects of disulfiram treatment on co-infection of bartonellosis with Lyme disease. In the study, 3 of 67 (4.5%) patients tested positive for both bartonellosis and babesiosis. In total, 10 of 12 (83.3%) patients co-infected with bartonellosis reported some form of improvement from at least one course of disulfiram treatment. The four patients receiving concurrent antimicrobial treatment, one on rifampin and minocycline and three on rifampin and azithromycin, did not achieve enduring remission yet experienced net benefit. However, the two patients who achieved enduring remission were on high doses of disulfiram without concurrent antimicrobial medications. Only one patient, who happened to have an immune deficiency, on high-dose disulfiram did not report any improvement after a 6-week treatment. Due to the small number of patients on concurrent antimicrobial treatment with disulfiram, we cannot conclude whether concurrent antimicrobial treatment is necessary for Lyme patients with bartonellosis co-infection; however, it appears that a high dose of disulfiram provided clinical improvement in Lyme patients with bartonellosis co-infection, independent of concurrent antimicrobial treatments.

Patients with babesiosis and bartonellosis co-infection tended to require higher doses of disulfiram to achieve clinical improvement (Table 4 and Table 5).

## 3. Discussion

We report on 67 individuals treated with disulfiram who were identified during a 3-year period from March 2017 to March 2020. Data were collected in the ordinary course of clinical care in one independent medical practice in the United States of America. Off-label use of FDA-approved drugs such as disulfiram, is within the discretion of practitioners and is not regarded as “experimental” nor does it require Institutional Review Board (IRB) approval [19]. Although there is great merit to formal research protocols and Randomized Controlled Trials (RCTs), there is increasing recognition of the value of Real World Evidence (RWE) and Real-World Data (RWD) as obtained through observational clinical studies. Berger et al. [20] attempt to define such studies by protocols that seem applicable to large vertically integrated healthcare systems equipped with electronic medical records, but they acknowledge this may not be suitable for all situations. Our observational report may be considered akin to a collection of “n-of-1” reports [21] since each of our subjects had had a track record of their courses using conventional antibiotics and often over very extended time periods. These patients often chose to pursue treatment with disulfiram in view of the limited effectiveness of their prior treatment and/or relapses after having derived benefit from antibiotics to a greater or lesser degree.

As for the mechanism by which disulfiram inhibits *B. burgdorferi*, it is known that disulfiram is an electrophile that is able to form disulfides with other thiol-bearing molecules [22]. *B. burgdorferi* has a variety of cofactors [23] containing thiophilic residues, which can be disrupted by disulfiram through thiol-disulfide exchanges. Disulfiram potentially causes the inhibition of *B. burgdorferi* metabolism as the formation of mixed disulfides with metal ions [24] would compete for the zinc and manganese cofactors that are crucial to the survival of *B. burgdorferi* [25]. A recent report of post-mortem study of one well-documented case of chronic and neurologic Lyme disease disclosed extensive borrelia-biofilm structures in all tissues studied (liver, kidney, heart, and brain), some as large as 300 microns in diameter. The persistence of borrelia DNA and antigens was confirmed by multiple methods [26]. The existence of biofilm is known to increase antibiotic resistance of resident microbes by some 1000-fold which may explain relapses in previously antibiotic-treated patients. In contrast to antibiotics, even the parent drug disulfiram, is a relatively small molecule likely to be more diffusible into biofilm than antibiotics. Carbon disulfide, a metabolite of disulfiram in humans in ng/L amounts [27], has antimicrobial properties, and its diameter in picometers is close to that of carbon dioxide which diffuses readily through metabolically active biofilm [28]. Although speculative, this might be one explanation for disulfiram’s apparent potency compared to conventional antibiotics in the treatment of Lyme disease. However, there are studies that disagree on the efficacy of disulfiram compared to other treatment options. In vitro, Alvarez-Manzo et al. [29] found disulfiram to be less effective than drugs like clarithromycin and nitroxoline at 5 µg/mL (16.9 µM), while Pothineni et al. found disulfiram be extremely effective at its minimum bactericidal concentration of 1.25 µM (0.371 µg/mL) [17], and Potula et al. found disulfiram to decrease in efficacy in concentrations greater than 5 µM (1.48 µg/mL). Together, these conclusions suggest that disulfiram becomes less effective at higher concentrations, which is supported by the fact that colloidal forms of disulfiram arise at higher concentrations [22]. Potula et al. also provide evidence from in vivo murine experiments that disulfiram was able to clear *B. burgdorferi* in 28 days post-infection, suppress cardiac inflammatory responses, and reduce the amount of antibody produced [22].

Disulfiram may also come with adverse effects, which have been well-characterized during its 70-year history as an FDA-approved agent for aversive treatment of alcoholism. Nonetheless, it continues to be utilized for that purpose. In 1990, Wright et al. [30] reported adverse reactions along with cardiac, hepatic, and neurologic toxicity at higher disulfiram doses of 250–500 mg/day. A monitoring schedule was suggested as a guideline for the providers’ supervision while their patients were taking disulfiram for the treatment of alcoholism. Similarly, more frequent monitoring measures were taken during the treatment of patients in our report considering this was a novel and off-label use of the drug.

Disulfiram is primarily metabolized in the liver through reduction, conjugation, and methylation to produce additional metabolites. A study on the elimination kinetics of disulfiram in alcoholics found a notable difference in the plasma levels of disulfiram and its metabolites among participants [31]. It took an average of 8 to 10 h for disulfiram and its metabolites to reach maximal plasma concentration, and their average half-lives ranged from 7.3 to 22.1 h. According to the study, variability in the metabolism of disulfiram may be the result of the lipid solubility of disulfiram, differences in plasma protein binding or the effect of enterohepatic cycling. This abundance of factors indicates a need for individualized disulfiram dosing strategies. Individuals with more rapid hepatic and renal clearance might need greater amounts of disulfiram to sustain benefits, while individuals with slower clearance could achieve the same benefits using lower doses over time.

Our suggested parameter of 4–5 mg/kg/day for high dose is aimed at a balance of benefit vs. risk but is not meant to be restrictive. Dosage must be individualized based on patient circumstances, responses and preferences. For example, Case 7 experienced remission on 7.1 mg/kg/day but relapsed. A subsequent course at 9.4 mg/kg/day resulted in severe, painful peripheral neuropathy at the 9-week mark that required cessation of disulfiram and moderate doses of gabapentin to control. Neuropathy subsided and resolved over months. However, the patient has enjoyed enduring remission, thus far lasting greater than one year following completion of her second course of treatment.

By examining the high-dose and low-dose patients, it can be inferred that the incidence of adverse reactions of disulfiram is proportional to its dose and to the total duration of treatment. Although irreversible injury has been reported, rarely, most adverse reactions are reversible with cessation of the agent, as proved to be the case in our relatively small group of patients who were closely monitored. A study conducted in 1984 by Christensen et al. [32] failed to show a difference between the rates of adverse reactions in alcoholic patients who took 200 mg/70 kg/day of disulfiram and those who took a placebo pill. This result is consistent with our findings in the low-dose group.

Recently, a retrospective study in France collected data through a standardized questionnaire of sixteen Lyme patients who had been treated with disulfiram [33]. In total, 13 of 16 (80%) patients reported toxic events, and 7 of 16 (44%) reported benefits for at least part of their symptoms. Major differences exist between this report and ours. The French study polled the “Lyme community” through a survey, seeking evidence of toxicity. The authors of the French study had no professional medical relationship with any of the respondents to their survey. The authors also had no documentation of the provenance of the patients other than that obtained through the questionnaires and had no way of verifying the accuracy of the responses. In contrast, this report is a retrospective review of individuals who had been personally cared for by two of the co-authors, examined, and followed closely through office visits and evaluated with initial thorough laboratory testing using a wide range of available validated methods. We also reviewed all pertinent past records and followed them through their treatments as well as inquired as to their ongoing status beyond completion of therapy or if remaining on therapy through July 2020. That the French study’s subjects reported both benefits and toxicities, but the title of their article cites only toxicities, raises questions as to the intent of the authors.

Furthermore, the toxicities of drugs are a relative matter, bearing in mind the seriousness of the illness being treated as well as the toxicities of other available therapies. Chronic and neurological Lyme disease is a very serious condition that can lead to death if untreated [3,26,34]. Intensive treatment is often necessary to prevent deterioration of the patient. Prior to the recognition of disulfiram’s possible utility, many clinicians and academicians have had to resort to intensive antimicrobial therapies that carry their own very well-documented suite of adverse and toxic reactions. By comparison, in our small cohort of patients followed closely and with judicious dosing, virtually all adverse effects were reversible and in no case was there any catastrophic complication nor fatal toxicities. We recognize the potential toxicity of disulfiram which has been well-reported in the worldwide peer-reviewed scientific literature in relation to its use for the treatment of alcoholism. We felt obliged in our May 2019 publication to bring these to the attention of treating clinicians so they could be mindful of these and vigilant to detect complications early in order to intervene (usually by suspending disulfiram) to try to avert serious injury.

Although the in vitro bench research finding of disulfiram to possess potency against Bb [17] and the favorable results from our prior [18] and the current clinical report are encouraging, more research is needed to look into the mechanisms of action for disulfiram and its metabolites. A recent peer-reviewed publication looked deeper into the mechanisms of disulfiram in a mouse model [22]. However, the amount of disulfiram these animals received intra-peritoneally far exceeded 4 mg/kg/day due to the limitations of oral absorption of the medication when given via the oral route or by gavage in mice. Translational research is needed to identify the mechanisms by which disulfiram and its metabolites exert salubrious or adverse effects when used in the treatment of Lyme disease in humans.

We are aware of the limitations of this report. Our data were collected solely from a clinical perspective and were not designed as a research study. As clinicians interacting with our patients, we rely heavily on what patients report to us, which we endeavored to faithfully reflect in the clinical record. We acknowledge the importance of placebo-effect as well as the possibility both that patients might have been influenced to exaggerate their degree of improvement in order to try to please the clinicians or that the clinicians unconsciously exaggerated the perceived benefit of the application of disulfiram on the patients reported here, despite best efforts to remain objective.

An IRB-approved clinical trial headed by Brian Fallon, M.D., M.P.H. at Columbia University’s Lyme and Tick-Borne Diseases Research Center is examining disulfiram as a test of symptom reduction among patients with previously treated Lyme disease [35]. The serial laboratory assessments and serial formal patient self-rating scales such as the General Symptom Questionnaire-30 (GSQ-30) [36] in the research protocol will provide more objective results. Additional studies, including multicenter, double-blind trials with comparator drug arms using serial patients rating scales, direct detection methods, and biomarkers could provide more objective, quantifiable and robust data.

Our experience as reported herein with well-characterized patients with Lyme disease who had pursued a relapsing course despite the prior application of more standard antimicrobial agents, indicates that disulfiram is a potent agent in the treatment of Lyme disease and may have activity against piroplasms as well and possibly bartonella. We do not regard disulfiram as a *panacea*, nor do we claim that we know it is a “cure” for these diseases. Nonetheless, the overwhelming majority of patients in whom it had been used perceived meaningful benefit, and some who pursued a high-dose approach enjoyed enduring remission, some for greater than one year, which had not been achieved with more conventional agents. Like any drug, its use does carry some degree of risk, which can be mitigated with judicious dosing and careful follow-up. In our opinion, it is a useful addition to the armamentarium of treatment for Lyme disease.

## 4. Materials and Methods

Inclusion criteria:History of Lyme disease diagnosed with supportive serologic findings, with or without detection of Lyme DNA by polymerase chain reaction;Prior provision of one or more courses of anti-microbial therapy for Lyme disease that either did not yield a favorable response or did yield favorable responses, but which was followed by a relapse of symptoms.

Exclusion criteria:Age less than 16 years old (due to lack of published data on safety);Alcohol use/abuse;Pregnancy;Psychosis;Rubber or latex allergy.

No normative data existed for the application of disulfiram in patients with Lyme disease prior to its use in the first three patients [18]. In those patients, a somewhat arbitrary dose of 500 mg/day was selected based on the upper limit of the dosage range commonly used for the aversive treatment of alcoholism. An arbitrary duration of 3 months was chosen. Case 2 in that series sustained a syncopal episode for which reason disulfiram was discontinued at 6 weeks after which the patient remained well for more than one year until sustaining a new deer tick bite. Each of the first three patients weighed in the vicinity of 95–100 kg so that, by chance, their dosages were in the 4–5 mg/kg/day range when reviewed retrospectively. Case 1, treated for some 3 months, has enjoyed enduring remission in excess of 3 years. Cases 1 and 2 roughly defined for us the lower and upper parameters of treatment duration once a “target” dose of 4–5 mg/kg/day was achieved (e.g., 6–12 weeks at “target” dose). We recognize this is still somewhat arbitrary and may not necessarily be “optimal” for any given patient. With the low-dose approach, we were surprised that extremely low doses (lowest in our series was 0.06 mg/kg/day) at or below 2 mg/kg/day still seemed to be salubrious. Due to disulfiram’s minimal toxicity at low doses and knowing that the drug has sometimes been used for years at a time for the aversive treatment of alcoholism, we prescribed it in a somewhat “open-ended” fashion for those in the low-dose cohort, although still continuing to follow patients carefully. Patients find this approach preferable to an open-ended antibiotic and anti-parasitic drug treatment approach (sometimes with multiple agents) which is more disruptive to the intestinal microbiome and entails significantly greater cost. Patients and their families and/or significant other(s) were informed about potential risks, potential benefits, and the considerable uncertainties involved in the application of disulfiram in the treatment of Lyme disease, and this was amply documented in the medical record; however, we did not require patients to sign an informed consent document.

It was routine to communicate formally in writing to the patient’s primary care physician and other practitioners important in the patient’s care in advance of applying disulfiram so that they could understand what we were doing and why and also so that they could communicate with us any concerns they had and/or whether they felt this course of action was ill-advised in our mutual patient. Direct communication was also important to elicit the cooperation of the patients’ other practitioners in the management of adverse reactions, should they occur.

In order to try to isolate the effect of disulfiram, cessation of any concurrent antimicrobial agents was required with the exception of those patients who had evidence of bartonellosis and who preferred to continue agents recognized as an effective therapy for that infection (e.g., tetracycline or an azalide with rifampin). Likewise, supplements and herbals were discontinued upon our advice in nearly all patients so as not to confound the effects of disulfiram and also because there exists limited data on disulfiram-herbal interactions as well as some limited data suggesting that some combinations might enhance the risk of hepatic injury [37].

While on disulfiram treatment, each patient had a monthly follow-up in person or by telephone to monitor their clinical status, review surveillance laboratory results, concurrently prescribed pharmaceuticals and any supplements or over-the-counter preparations.

## 5. Conclusions

In our experience, disulfiram monotherapy is useful in the treatment of Lyme disease. Regular laboratory monitoring and close clinical follow-up are necessary. Dosages of 4–5 mg/kg/day for 6–12 weeks appear to be optimal for attempting to achieve enduring remission while minimizing adverse effects. Dosages as low as 0.06–2 mg/kg/day for indeterminate durations also conferred benefit with minimal adverse effects. An individualized and flexible approach with shared decision-making with patients is particularly suitable in the use of this agent. Further study of disulfiram at academic research centers to clarify its mechanisms of action and to conduct formal treatment trials seems warranted.

## Figures and Tables

**Table 1 antibiotics-09-00868-t001:** Summary of patients on high dose (≥ 4 mg/kg/day) disulfiram (*n* = 33).

Patient ID	M/F	Age	Weight (kg)	Max Dose (mg/kg/day)	Duration	Infection & Other Conditions	Other Rx	Adverse Reaction	Benefit
**1**	M	29	95	5.3	6 WK	Bb		Psych	Y E
**2**	F	24	57	5.5	5 MO	Bb, Bh		PN, FAT	Y
**3**	F	61	71	7 7	6 WK 4 WK	Bb		PN, Psych, syncope	Y Y E
**4**	F	64	82	4.6	5 MO	Bb		FAT, Psych, Arth, LFT	Y
**5**	M	59	64	5.8	6 WK	Bb, Bm		FAT	Y
**6**	M	29	65	7.7	5 MO	Bb, CRPS		PN, FAT, Arth	Y
**7**	F	27	106	7.1 9.4	6 WK 9 WK	Bb, Bd		PN, FAT, Arth	Y Y E
**8**	F	62	60	8	11 WK	Bb		FAT, Psych, LFT, near syncope	Y E
**9**	F	63	70	5.4	3 MO	Bb, Bm		FAT, hyperdynamic	Y
**10**	M	59	100	3.3 5.0	4 MO 3 MO	Bb, Bm		FAT	Y Y E
**11**	M	50	88	5.7 5.7	6 WK 12 WK	Bb, RMSF		FAT	Y E Y E
**12**	F	66	54	9.3	14 WK	Bb, Bm		PN, FAT, Psych	Y E
**13**	M	70	73	6.8	3 MO	Bb		PN, FAT, Psych	Y
**15**	F	55	60	4.1	2 MO	Bb, Bm		PN, FAT, LFT	Y E
**16**	M	21	60	8.3	12 WK	Bb, Bm, Bh		Psych, FAT, LFT	Y E
**17**	M	52	103	7.3	10 WK	Bb, Bm		Psych	Y
**18**	M	50	93	5.4	3 MO	Bb, Bm		Psych	Y E
**19**	F	46	55	6.8	10 WK	Bb, Bh	azith, rifam	FAT	Y
**20**	F	34	60	6.25	6 MO	Bb, OCD		FAT, Psych	unclear
**21**	F	62	75	6.7 6.7	6 WK 3 MO	Bb, Bh		FAT	Y Y E
**22**	M	28	59	6.4	6 WK	Bb, Bh, Imm Def			N
**23**	F	63	34	7.1 3.6	2 MO 4 MO	Bb		FAT, PN Psych	Y Y
**24**	F	48	64	4.9	8 MO	Bb, Bm, POTS		Psych, FAT, Arth	Y
**29**	F	23	59	5.3	6 MO	Bb		FAT, Psych after d/c	Y
**34**	F	51	59	6.4	7 WK	Bb		FAT, Arth	Y
**36**	F	18	77	4.8	7 MO+	Bb		PN	Y
**37**	F	74	115	6.5 6.5	6 WK 3 MO+ (alt. mos. on & off)	Bb			Y Y
**38**	M	71	95	5.3	4 MO	Bb		LFT	Y
**39**	F	59	63	6 2	5 MO 4 MO+	Bb, Bh	azith+, rifam	FAT, Arth, Psych	N Y
**42**	M	25	92	5.4	4 MO	Bb, Bd, TBRF		Psych	Y
**49**	M	83	76	5.8 1.6	6 WK 2 MO+	Bb, Bm, myopathy		syncope	Y E Y re-bitten
**50**	F	37	66	7.5 3.8 0.6 0.6	6 WK 3 WK 2 WK 3 MO+	Bb, Bm		Psych, FAT	N N Y Y
**54**	M	73	85	4.4	4 MO+	Bb, Bm, Bh	mino, rifam		Y re-bitten

**Table 2 antibiotics-09-00868-t002:** Summary of patients on low dose (< 4 mg/kg/day) disulfiram (*n* = 34).

Patient ID	M/F	Age	Weight (kg)	Max Dose (mg/kg/day)	Duration	Infection & Other Conditions	Other Rx	Adverse Reaction	Benefit
**14 ^1^**	M	61	76	1.6	3 MO	Bb, Bm, MND		LFT	Y: constitution N: neurologic
**25**	F	17	59	2.1	3 MO	Bb		LFT, Psych	Y
**26**	M	71	85	2.9	2 MO	Bb, TBRF, MSA-C		Psych	N
**27**	F	60	64	2	4 MO	Bb		FAT	Y
**28**	F	52	64	2.9	4 MO	Bb, MCAS			Y
**30**	M	34	79	3.1	4 MO	Bb, Bh			Y
**31**	M	36	61	0.3	4 MO	Bb		FAT	Y
**32**	M	20	66	0.5	6 MO	Bb			unclear
**33**	F	55	57	0.9	7 MO	Bb, Bh			Y
**35**	M	64	68	0.6 0.6	5 MO 4 WK	Bb, TBRF			Y, marginal
**40**	F	58	50	1.3	4 MO+	Bb			Y
**41**	M	21	91	0.9	8 WK	Bb, Bh			unclear
**43**	F	20	48	0.3	6 MO+	Bb		FAT, Arth	Y
**44**	F	57	70	0.6	3 MO+	Bb		FAT	Y
**45**	F	72	106	0.6	5 MO+	Bb		FAT	Y
**46**	M	17	159	1.5	8 MO+	Bb, Bd		FAT, LFT	Y
**47**	F	19	104	0.1	3 MO+	Bb			Y
**48**	F	64	66	0.9 0.9	4 MO 3 MO+	Bb, Bh		FAT, LFT	Y Y
**51**	M	83	99	1.1	4 MO+	Bb, OBS		FAT	Y
**52**	F	64	68	0.5	2 MO+	Bb			Y
**53**	F	23	95	1.6	7 MO	Bb, Bh		PN, Arth	Y
**55**	M	17	84	1.5	2 MO+	Bb, Bh	azith, rifam		Y
**56**	M	39	83	0.2	5 MO+	Bb		FAT, Psych	Y
**57**	M	61	80	0.1	1 MO+	Bb, NMD			Y
**58**	M	19	86	0.2	6 MO+	Bb		FAT, LFT	Y
**59**	M	79	120	0.3	11 MO+	Bb, MG-like NMD	pyridosti- gmine		Y
**60**	F	18	54	0.8	3 MO+	Bb, Bd, TBRF			Y
**61**	M	30	73	3.4	5 MO+	Bb			Y
**62**	F	51	43	1	16 MO+	Bb, Bm			Y
**63**	M	69	113	0.6	2 MO+	Bb		Psych	Y
**64**	M	59	105	1.2	2 MO+	Bb			Y
**65**	F	43	49	2.5 0.4	4 MO 4 MO+	Bb, Bm, Bh		FAT, Psych	Y Y
**66**	F	18	57	0.3	2 MO+	Bb	clarithro		Y
**67**	M	39	80	3	6 WK	Bb			Y

^1^ Patient deceased from MND.

**Table 3 antibiotics-09-00868-t003:** Risk-benefit comparison between the two disulfiram treatment styles.

		High Dose Group (*n* = 33)		Low Dose Group (*n* = 34)		*p*-Value ^1^	Significance
**Benefits**	Symptomatic Improvement	31	93.9%	31	91.2%	0.66701	No
	Enduring Remission (> 6 months)	12	36.4%	0	0%	0.00010	Yes
**Adverse Reactions**	Fatigue (FAT)	22	66.7%	11	32.4%	0.00497	Yes
	Peripheral neuropathy (PN)	9	27.3%	1	2.9%	0.00520	Yes
	Psychiatric Symptoms (Psych)	16	48.5%	5	14.7%	0.00288	Yes
	Elevated LFT	5	15.2%	5	14.7%	0.95918	No

^1^*p* < 0.05 is defined as statistically significant using the χ^2^ test.

**Table 4 antibiotics-09-00868-t004:** Effects of disulfiram treatment on babesiosis co-infection of Lyme disease (*n* = 15).

Patient ID	M/F	Age	Weight (kg)	Max Dose (mg/kg/day)	Duration	Infection & Other Conditions	Other Rx	Adverse Reaction	Benefit
**5**	M	59	64	5.8	6 WK	Bb, Bm		FAT	Y
**7**	F	27	106	7.1 9.4	6 WK 9 WK	Bb, Bd		PN, FAT, Arth	Y Y E
**10**	M	59	100	3.3 5.0	4 MO 3 MO	Bb, Bm		FAT	Y Y E
**12**	F	66	54	9.3	14 WK	Bb, Bm		PN, FAT, Psych	Y E
**14 ^1^**	M	61	76	1.6	3 MO	Bb, Bm, MND		LFT	Y: constitution N: neurologic
**15**	F	55	60	4.1	2 MO	Bb, Bm		PN, FAT, LFT	Y E
**17**	M	52	103	4.9	10 WK	Bb, Bm		Psych	Y
**18**	M	50	93	5.4	3 MO	Bb, Bm		Psych	Y E
**24**	F	48	64	4.9	8 MO	Bb, Bm, POTS		Psych, FAT, Arth	Y
**42**	M	25	92	5.4	4 MO	Bb, Bd, TBRF		Psych	Y
**46**	M	17	159	1.5	8 MO+	Bb, Bd		FAT, LFT	Y
**49**	M	83	76	5.8 1.6	6 WK 3 MO+	Bb, Bm, myopathy		syncope	Y Y re-bitten
**50**	F	37	66	7.5 3.8 0.6 0.6	6 WK 3 WK 2 WK 3 MO+	Bb, Bm		Psych, FAT	N N Y Y
**60**	F	18	54	0.8	12 WK	Bb, Bd, TBRF			Y
**62**	F	51	43	1	16 MO+	Bb, Bm			Y

^1^ Patient deceased from MND.

**Table 5 antibiotics-09-00868-t005:** Effects of disulfiram treatment on bartonella co-infection of Lyme disease (*n* = 12).

Patient ID	M/F	Age	Weight (kg)	Max Dose (mg/kg/day)	Duration	Infection & Other Conditions	Other Rx	Adverse Reaction	Benefit
**2**	F	24	57	5.5	5 MO	Bb, Bh		PN, FAT	Y
**16**	M	21	60	8.3	12 WK	Bb, Bm, Bh		Psych, FAT, LFT	Y E
**19**	F	46	55	6.8	10 WK	Bb, Bh	azith, rifam	FAT	Y
**21**	F	62	75	6.7 6.7	6 WK 3 MO	Bb, Bh		FAT	Y Y E
**22**	M	28	59	6.4	*6 WK*	Bb, Bh, Imm Def			N
**39**	F	59	63	6 2	5 MO 4 MO+	Bb, Bh	azith+, rifam	FAT, Arth, Psych	N Y
**41**	M	21	91	0.9	8 WK	Bb, Bh			unclear
**48**	F	64	66	0.9 0.9	4 MO 3 MO+	Bb, Bh		FAT, LFT	Y Y
**53**	F	23	95	1.6	7 MO	Bb, Bh		PN, Arth	Y
**54**	M	73	85	4.4	4 MO+	Bb, Bm, Bh	mino, rifam		Y re-bitten
**55**	M	17	84	1.5	2 MO+	Bb, Bh	azith, rifam		Y
**65**	F	43	49	2.5 0.4	4 MO 4 MO+	Bb, Bm, Bh		FAT, Psych	Y Y

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
