# Peer review of "“Repurposing” Disulfiram in the Treatment of Lyme Disease and Babesiosis: Retrospective Review of First 3 Years’ Experience in One Medical Practice"

_antibiotics, 2020, doi:10.3390/antibiotics9120868_

Round 1

Reviewer 1 Report

The manuscript by Gao et al., aims at providing a study showing the possible effects in treating symptoms of Lyme disease (and Babesiosis) using Disulfiram. This is not a clinical study , there are no controls/no placebos and the authors appear to be providing their data based on no standardised methods when prescribing disulfarim or the period of time that it should be taken.

  1. They divided their patients into two groups of low and high dose for disulfiram but with no real explanation of what differentiated these two groups into being labelled as low and high besides the amount of disulfiram. More clear criteria for dividing patients needs to be given .
  2. They write about courses of high or low dose treatment but I cannot seem to find the duration of a particular course, the amount of time between courses and if there were changes to certain courses.
  3. The Introduction for the article is particularly short and seems to be very anecdotal in citing an article upon which this study is based. Stating how many time a publication is downloaded or been discussed on Facebook suggests, the authors are trying to bring more attention to the article by Kenneth Liegner.
  4. The conclusions of this study are extremely extrapolated especially given that there were no control patients and suggesting at this time that this therapy is useful in treating Lyme disease and other tick-borne diseases holds no scientific weight.
  5. There is also a government funded clinical trial which does have controls which the authors seemed to have omitted for mentioning https://clinicaltrials.gov/ct2/show/NCT03891667). 
  6. Most importantly the number of patients that have side-effects associated with taking disulfiram is the larger majority, compared to those who say they have recovered. This sets a dangerous precedent and unless the authors can provide controls working with similar patients  who are given placebos, this study cannot be considered to be acceptable in its current form

Author Response

Manuscript ID antibiotics-908951

Responses to Reviewer 1

Reviewer 1 is correct in concluding the mss. does not report on a ‘clinical study’.There are no control and placebo groups. Also correct is the observation that the data is based on no ‘standardized methods’ when prescribing disulfiram including dosages and durations of treatment.

This is so because, other than the Liegner publication in MDPI-Antibiotics of May 2019(Liegner, K.B. Disulfiram (Tetraethylthiuram Disulfide) in the Treatment of LymeDisease and Babesiosis: Report of Experience in Three Cases.Antibiotics2019,8, 72.)there exists no precedent and no normative data for the use of disulfiram in the treatment of persons with Lyme disease in the medical literature.

The dosages reportedin that article were somewhat arbitrarily chosen as the upper limit of customary doses of disulfiram used for the treatment of alcoholism(500 mg/day) because this dosage had proven to be generally safe and tolerated in that patient population (e.g. chronic alcoholics).

 As it turned out in the three cases reported on each of whom weighed in the vicinity of 200 pounds or some 91 kgtheir doses approximated, onretrospective review,4-5 mg/kg/day.

Likewise, duration of treatment in those three patients was somewhat arbitrary as there was no data whatsoever as to what might be a ‘necessary’ or ‘sufficient’ duration of treatment with disulfiram in persons suffering from Lyme disease. The author of that paper, again, somewhat arbitrarily aimed at approximately3 months duration of treatmentbecause that duration between visits had become usual during the care of Case 1 when he had been on long-term antimicrobial treatment. As it happened, one patient(Case 2 in that publication) completed only 6 weeks at full dosage of 500 mg/day when he experienced a syncopal episode requiring its discontinuance. It was observed thatthis patient experienced an enduring remission of his previously extremely debilitating symptoms of Lyme disease and babesiosis (which had been recurrent) lasting more than one year and until he sustained a new deer tick bite. So, this experience led to the conclusion that some persons with even serious chronic and neurologic Lyme disease could be treated with as little as 6 weeks with disulfiramat relatively high dosageand remain clinically well. This defined for us the lower endof treatment durationwhen we used the high dosage approach. Case 1, who was treated for 3 months with disulfiram, 500 mg/day has remained clinically wellin excess of 3 years. Previously he had pursued a relapsing course whenever intensive antibiotics were discontinued.He did require a psychiatric hospitalization but it remains unclear whether or not disulfiram was causative since he was under a great deal of situational stress at the time.Our experiencewith Case 1 and Case 2served as our basis for selecting 6 weeks and 12 weeks as the parameters for the low end and high end of duration of dosing once a rough ‘target’dose of 4-5 mg/kg/day was achieved.

So, the 3 cases reported in the May 2019 publication were the first humans treated with disulfiram for Lyme disease. This experiential clinical data was the onlydata available to extrapolate from.No normative data hadexisted previously for application of disulfiram in patients withLyme disease.

We were extremely impressed with the effectiveness of disulfiram in the subset of patients with whom we have been dealing: those with chronic Lyme disease for whom, until the identification of possible utility of disulfiram based on in vitrowork by Drs. Pothineni and colleagues, antibiotics had been the mainstay of treatment. Although antibiotic therapy has been helpful in yielding improved quality of life for many such patients, it became apparent to us and many other workers that antibiotics frequently were suppressive and that many patients relapsed whenantibiotics were withheld, a problematic situation.

Encouraged by the favorable outcomes in those first 3 cases, disulfiram was offered as one option for patients deemed suitable and, as reported 71 persons opted for this approach during the 3 year periodfrom March 2017 through March 2020. Four individuals were ‘lost to follow-up’leaving67 evaluable patients.

This was not a ‘study’. This was not a ‘research project’. This was not an Institutional Review Board (IRB)-approved research study nor was IRB-approval necessary or required. In the United States and specifically within New York State, off-label use of previously Food and Drug Administration(FDA)-approved medications is within the prerogative of the treating physician and does not require any regulatory approval. Although it may be ‘novel’, it is not considered ‘experimental’ nor ‘research’ per se.

Although the authors of the 2019 study and the current study maintain hospital affiliations none is a hospital nor a government employee and the conduct of their practice is independent of their affiliated hospitals and publicationswhich ensueon experiences derived from the ordinary course of clinical practice do not require ‘pre-review’ by the affiliated hospital or hospital systems. If a research studywere truly undertaken,IRB-approval would be required.This was not the case in the current work reported in MDPI-Antibiotic ID 908951.

Whereas there is great merit in controlled blinded studies with comparator arms and/or placebo controls, such studiestypically compare one type of treatment with another type of treatment in order to compare both outcomes and adverse effects.

The experience we reporton, which was in essence ‘uncharted territory’,does not comprise such a study nor was it intended to beso.On the other hand, each patient of the 67 reported upon isakin to an“N of 1”trialwhere there pre-disulfiram status might be considered to be the ‘comparator’ to their post-disulfiram status(Lillie EO, Patay B, Diamant J, Issell B, Topol EJ, Schork NJ. The n-of-1 clinical trial: the ultimate strategy for individualizing medicine?.Per Med. 2011;8(2):161-173. doi:10.2217/pme.11.7).Virtually all of them had reported similar experience of months or years of illness with round-robin courses of various types of antibiotic treatment –yes with benefit –but with invariable and demoralizing relapses.

Quoting from the Lillie article:

“An intuitive way around this is to treat the individual patient as a study subject and objectively and empirically determine the best course of therapy. Such single subject or ‘n of 1’ trials have great precedent in educational and behavioral settings, but have not been used to an appreciable degree within the medical and clinical communities; in fact, many such trials have been disparaged as ‘only anecdotal’ [20]. There are many reasons for this, not least of which is cost, but n-of-1 studies are a promising way to advance individualized medicine and a method for gaining insights into comparative treatment effectiveness among a wide variety of patients.”

And later, in the same article:

“...In this light, n-of-1 trials that focus exclusively on the objective, empirically determined optimal intervention for a single patient or subject clearly defy easy generalizability, but are compatible with the ultimate end point of clinical practice –the care of individual patients. In addition, clinical studies focusing on the treatment of single patients is, as noted previously, actually more consistent with the vision of individualized or personalized medicine than stratifying patients in to groupsmore or less likely to benefit from a specific treatment on the basis of population-level association studies [24,25]. Finally, as discussed later, n-of-1 trials could be very efficientand less costly vehicles for motivating serious consideration about an intervention with respect to other patients, larger patient groups, or other clinical conditions.”

We also bring Reviewer 1’s attention to the increased recognition of thevalue of observational reports as a complement to Randomized Controlled Trials:Berger, M.L.;Sox, H.;Willke, R.J.; Brixner, D.L.; Eichler, H.; Goettsch, W.; Madigan, D.; Makady, A.; Schneeweiss, S.; Tarricone, R.; et al. Good practices for real-world data studies of treatment and/or comparative effectiveness: Recommendations from the joint ISPOR-ISPE Special Task Force on real-world evidence in health care decision making. Pharmacoepidemiol. Drug Saf. 2017, 26, 1033—1039, doi:10.1002/pds.4297.Although the formalization of protocols they argue for are most suitable for large vertically-integrated healthcare systems, they acknowledge that this may not be suitable for all situations.

During the course of working with subsequent patients it became clear that dosagesneeded to be adjusted to the individual and a ‘standard’ dose was not appropriate.As mentioned above, in the original article a somewhat arbitrary dose was selected since there was no prior experience with applying disufiram in the treatment of Lyme disease. Some patients requested much more cautious dosing for a variety of reasons including pre-existing neuropathy (e.g. from Lyme disease)wishing to minimize the risk of induction of a superimposed drug-induced neuropathy. Others had such a precarious clinical state that joint decisions were made to proceed with very modest doses. Also, it became clear that disulfiram provoked in some individuals what was deemed to be

profound Jarisch-Herxheimer-like reactions, once again requiring a measured and cautious approach. Other patients sought and tolerated more intensive treatment and were influenced to pursue more aggressive treatment in view of the enduring remissions that had been obtained with the (relatively high) doses reported in the first 3 casesfrom the May 2019 article. It was only upon retrospective analysis of our experience was it realized that 4 mg/kg/day was a suitable dividing line between ‘high dose’ and ‘low dose’ regimens and the characteristics of the patients and the effects of treatments at high and low doses displayed different side effect profilesalthough both approaches yielded benefit. It is not possible (thus far) to determine whether the low dose approach can yield enduring remissions whereas a significant proportion of those pursuing the high dose approach were able to enjoy such a status. Again, the report is observational and experiential in nature and we did notaimto define high and low dose approachesat the outset. However, our 3 years’ experience revealed this type ofa ‘break-out’ among the persons treated with disulfiram.

Response to specific numbered points from Reviewer 1:

1.A detailed review and recitation of each of the 67 cases, we feel would compriseadditional lengthy verbiage –reportage of the clinical history of our Cases 1-3 in our original article required 10+ pages in the final published version. We hope the editor, publisher and reviewers can grant, based on our background and experience in this field, that the cases ‘qualified’ as bona fidecases of persons with Lyme disease with supportable histories and with reasonable laboratory corroboration. Although we could provide detailed documentation for each of the 67 cases, this will be a time consuming task and would require likely months-long delay in possible publication of the current mss.,the substance of which, we feel, is timely and ‘time-sensitive’. Any mss. can be improved –and in fact we are modifying the mss. based on comments of all reviewers -we believe a detailed precis’of each case isunnecessary.

2.We agree this remains an anecdotal area as of this point. The original report concerned only 3 individuals, but so striking was the responseto disulfiramin these persons(cases whichhad been so protracted and had requiredextraordinarily intensive anti-microbial treatment in order to ‘keep their heads above water)that thismotivated the author to undertake the arduous process of publication of this experience.The current report describes an additional 67 patients, which is still a very limited number, and yes, we would agree it still could be described as ‘anecdotal’. We call for additional studies including formal research studies that would be most appropriately undertaken by academicians and which would include objective biomarkers, serial direct detection methods and validated rating scales of patient status, e.g. more ‘robust’ data.

3.We disagree thoroughly with the assertion of Reviewer 1 here. Controlled double-blinded studies are far from the sole source of truth in medical matters. The authors are very experienced clinicians and the guiding author has been involved with diagnosis and treatment of Lyme disease and other tick-borne diseases since the late 1980s, was a memberof the Treatment Panelin the 1991 State-of-the-Art Conference on Lyme Disease convened in Bethesda, Maryland by the National Institute of Allergy and Infectious Diseases (NIAID) and the National Institute of Arthritis and

Musculosketal Diseases (NIAMSD)–Clinical Courier, Vol. 9, No. 5 August 1991 ISSN 0264-6684NIAMS & NIAID–see PDF-has presented at national and international conferences on Lyme disease and has authored numerous peer-reviewed published articles on complex cases of Lyme disease including serious and fatal cases of neurologic illness(Liegner KB, Duray P, Agricola M, Rosenkilde C, Yannuzzi L, Ziska M, Tilton R, Hulinska D, Hubbard J, Fallon B. Lyme Disease and the Clinical Spectrum of Antibiotic-Responsive Chronic Meningoencephalomyelitides. J Spirochetal and Tick-borne Dis 1997;4:61-73.See attached PDF).Careful longitudinal studies of cases are extremely important and have added greatly to medical knowledge (see Curriculum vitae of authors). Who can say that clinical observationsof individual cases by diligent practicing physicians are not valuable? Was not William Beaumont’s careful observation of Alexis St.Martin, afforded by serendipitous circumstance (not too unlike the serendipitous circumstancesthat led to the first application of disulfiram in humans)of enormous importance for medical and scientific understanding of human digestion(Markel H. Experiments and observations: howWilliam Beaumont and Alexis St. Martin seized the moment of scientific progress. JAMA. 2009Aug 19;302(7):804-6. doi: 10.1001/jama.2009.1212. Erratum in: JAMA. 2009 Oct 7;302(13):1420.PMID: 19690317).

Again, we ask Reviewer 1 toconsider the Lillie and Berger articles referred to above, regarding the value of observational studies.

5.Point well-taken. We will be sure to mention the Clinical Trial alluded to by Reviewer 1, however, it is not a ‘government-funded’ trial, although it is listed on a government web-site (Clinicaltrials.gov). The Fallon trial is funded through private sources. Due to COVID-19 the trial is ‘on hold’. I understand that Dr. Fallon is seeking to modify the study protocol so that it can incorporate at least some ‘telemedicine’ in order to facilitate conduct of the trial (personal communication with Dr. Fallon).

6.The original publication of May 2019 included fairly detailed review of potential adverse effects of disulfiram usage which had been reported in the literature of its use for alcoholism for the past 70 years or so. It is worth mentioning(and perhaps including in a revised manuscript) that early on in the history of disufiram doses of 1 to 3 grams/day were commonplace. Perhaps that may explain someof the more serious and irreversible injuries reported in the literature. Also, it would not be surprising if persons suffering from chronic alcoholism might have had less than ideal or regular clinical follow-up.We have been forthright in reporting the adverse effects of treatment in our patients, both high dose and low dose. However, such adverse effects werereversiblein all but a few patients (whose peripheral neuropathies have partially but not fully reversed) and the overwhelming majority of patients were able to endorse a ‘net’ benefit of treatment. Even if they did experience adverse effects, be it temporary emotional upset or painful neuropathy, upon questioning they have nearly all said that they derived a ‘net’ benefit and stillwould havechosen disulfiram. For almost all of the patients disulfiram use was an overall beneficial experience as in Case 1. reported in the original article when we discussed the possible role of disulfiram in his need for psychiatric hospitalization, he was adamant: “It was worth itand don’t failto offer it to patients just because of THAT!”. One also has to weigh the usually reversible adverse effects of disulfiram in the treatment of chronic and neurologic Lyme disease with the potential serious and sometimes irreversible adverse effects of use of intensive antibiotics for the same condition (e.g. potentially fatal line sepsisandC. difficilecolitis, gall bladder complications etc.) not to mention the enormous expense of intravenous antibiotic therapy and (for example) atovaquone at $1000/bottle for months on end. Disulfiram’s cost, by comparison is very modest in dollar amountand risks and toxicities are also quite modest in comparison to the types of prolonged and intensive antimicrobial treatments that most practitioners whodeal with this type of illness have often found necessary. We maintain that the experience reported in the mss. has intrinsic value and will constitute a worthwhile addition to the worldwide peer-reviewed medical and scientific literature. We hope that it will stimulate further studies by others to define disulfiram’s mechanisms of action, optimal use andproper role as one tool in the armamentarium of treatments for Lyme disease. We do not claim it is a ‘panacea’ for Lyme disease and we make no ‘claim’ of it being a ‘miracle cure’. It is potent. It is not an antibioticin the usual sense.We have not seen any cases of C. difficileresulting from its use and in our opinion routine probiotic usage is unnecessary.We have witnessed many persons benefitting from it and we have endeavored to show how disulfiram can be used responsibly with relative safety and effectiveness, which has been the experience in our practice over the past 3+ years.

Reviewer 2 Report

Manuscript Number: Antibiotics-908951

Title: Repurposing Disulfiram in the Treatment of Lyme Disease and Babesiosis: Retrospective Review of First 3 Years’ Experience in One Medical Practice

Article Type: Article

Comments to the authors:

Summary: The paper evaluates the benefits and adverse reactions of the treatment of Lyme disease using Disulfiram in 67 patients over a period of three years. I highly recommend the publication of this article in Antibiotics after consideration of a few minor comments below:

  1. Introduction:

L.30 more potent against Borrelia burgdorfei (against instead of versus)

The introduction is very short. I think it would help the readers to know what kind of drug disulfiram is? What is it normally used for? Perhaps move part of the description from discussion (L.116-128) to introduction. I do not think it is appropriate to mention the number of downloads or facebook memberships in a scientific report. In fact, it would help to summarize what the article was about in the high throughput screening.

It would also help to write a paragraph or two about Lyme disease and what are the other conventional treatments followed for this disease. This should be described first followed by a short description about disulfiram.

  1. Was there no requirement for ant IRB or IEC approval of the human subjects research, including the protocol, informed consent document etc for this study?

Did they perform any serological tests to confirm Ag elimination? If not and they only looked for improved symptoms, this needs to be mentioned.

It is good that the authors warn the readers about not carrying out a double-blind placebo study towards the end.

I find the results encouraging and a good starting point for further research and wish the authors all the best for a systematic study at Columbia University.

Author Response

Manuscript ID antibiotics-908951

Responses to Reviewer 2

The authors deeply appreciate the general support for the mss. expressed by Reviewer 2.

1.Introduction -disulfiram was more potent in vitro than doxycycline in thePothineni article in Drug Development, Design & Therapy.

The authors assumed, perhaps unjustifiably,readers of the current mss. if published would already have read the MDPI-Antibiotics article of May 2019 (Liegner, K.B. Disulfiram (Tetraethylthiuram Disulfide) in the Treatment of Lyme Disease and Babesiosis: Report of Experience in Three Cases.Antibiotics2019,8, 72.)in which much of the background information to which Reviewer 2alludes, was discussed in depth. However, as each mss. and article ought to ‘stand on its own’ it would not hurt to briefly reiterate some of those key elements in order to familiarize readers, including those who may not have seen or read the May 2019 article. The authors will revise the introduction in order to address Reviewer2’s suggestions.We are unsure whether re-locating lines 116-128 in to an introduction would be optimal for the introduction; but agree a fuller introduction would benefit the mss.

1.We do not agree it is inappropriate to point out the number of downloads(which now exceed 20,000) or the extensive interest amongst patients as evidenced bymembership numbers for disulfiram-dedicated Facebookgroups since these areverifiable factual datawhichspeak to thirst among patients and practitioners for additional options for treatment of Lyme disease above and beyond antibiotics alone. Decades ofdenial by some academicians, some government agencies and some in the pharmaceutical industry of the reality of Lyme disease as a potentially chronic infectionhas stagnated progress in the development of improvements in therapeutics. One of the early opinion pieces concerning Lyme disease published in Scienceindicates by its titleand contentthat Lyme disease is both a biological as well as asociologicalphenomenon (Barbour AG, Fish D. The Biologic and Social Phenomenon of Lyme Disease. Science 1993;Vol 260: pp.1610-1616 –copy attached). The download numbers and Facebook membership numbers corroborates that point of view.These data were not cited from the authors’ point of view in any way to ‘pump up’ or ‘promote’ the author or the May 2019 article nor the authors of the mss. presently under consideration.Despite our disagreement,out ofrespect of Reviewer 1’s feelings, we have deleted mention of the number of downloads orthe number of members of disulfiram-focused Facebook groups.

2.In the United States and specifically within New York State, off-label use of previously Food and Drug Administration (FDA)-approved medications is within the prerogative of the treating physician and does not require any regulatory approval and, although it may be ‘novel’, it is not considered ‘experimental’ nor ‘research’ per se. This was not a ‘study’. This wasnot a ‘research project’. This was not an Institutional Review Board (IRB)-approved research study nor was IRB-approval necessary or required. Although the authors of the 2019 study and

the current study maintain hospital affiliations none is a hospitalor a government employee and the conduct of their practice is independent of their affiliated hospitals. Publications which ensue on experiences derived from the ordinary course of clinical practice do not require ‘pre-review’ by the affiliated hospital or hospital systems. Were they truly ‘research’ IRB-approval would be required.. This was not the case in the work reported in MDPI-Antibiotic ID 908951.We never sought to obtain anysigned informed consents for use of disulfiram from any of the patients treated in the 2019 publication nor in those reported in the current retrospective report. However, each was fully informed about potential risks, potential benefits as well as uncertainties about the use of disulfiram in the setting of Lyme and other tick-borne disease and this is amply reflected in the medical record of each patient. Additionally, prior to employing disulfiram, it was our practice to correspond in writing to each of the other physicians or practitioners with significant involvement inthe patient’s care. Each patient received a copy of the letter and this also served to document careful informing of the patient. This is mentioned in lines 244-253 of the current mss. We have subsequently learned that some practitioners have opted to require patients to sign a formal informed consent prior to use of disulfiram. We have not done so. We did not routinely obtain serial serologies not direct antigen detection testing (e.g. polymerase chain reaction [PCR] tests for detection of Lyme disease, babesiosis or bartonellosis or fluorescent-in-situ hybridization testing [F.I.S.H.] for babesiosis or bartonellosis) because this would have entailed added and often unnecessary expense for patients who would have had to bear this expense in the event such testing was not covered by insurance. As many of the patients have enjoyed very improved quality of life after or while undergoing disulfiram treatment, they had little motivation to pursue serial testing. One exception is Case 3 of the original report who experienced symptoms of relapse and who requested testing which, in fact, revealed a positive Lyme PCR which correlated with his clinical status. Surely, funded research project ought to incorporate serial testing including serologies, direct detection methods and any useful biomarkers or indirect indicators of ongoing infection of inflammation. As clinicians caring for ill persons, we did not feel it would be appropriate to insist on such testingif patients perceived no need for it in order to determine their care. The clinical trial under the auspices of Columbia University is presently ‘on hold’ due to COVID-19. I understand from Dr. Fallon that he is seeking a modification of his protocol to allow tele-medicine so that the trial can resume.We believe serial laboratory assessments and serial formal validated patient self-rating scales are a part of that research protocol.

Reviewer 3 Report

General Comments

The manuscript by Gao and colleagues entitled "Repurposing Disulfiram in the Treatment of Lyme Disease and Babesiosis: Retrospective Review of First 3 Years’ Experience in One Medical Practice" examines the effect of disulfiram (DSF) in 67 patients with Lyme disease and coinfections. The study is certainly significant for the Lyme community that has developed a fixation on DSF, but more definition and detail are needed to flesh out the study results.

Major Comments

1. The Title and Abstract describe the study as a "retrospective review" of patient experience with DSF, yet in the Discussion on page 6 the authors criticize a less favorable French study of DSF because it "was conducted in a retrospective manner, which generally increases the likelihood for recall bias." It is unclear how the studies differ if they are both retrospective, and the criticism suggests that the current study is also open to recall bias.

2. The patient population in the study requires further characterization. In Materials and Methods on page 11, the stated inclusion criteria are simply a positive Lyme test and failure of at least one course of antibiotic therapy. The length of prior illness, quality of life (QOL) measures and clinical symptoms of patients, type and length of prior treatment and specific coinfection treatments are not stated. It appears that all patients suffered from chronic Lyme disease (CLD), and a definition of this diagnostic entity should be provided (Stricker & Fesler, Am J Infect Dis 2018; 14:1-44). It would help to have a Table listing all symptoms in order to compare the high-dose and low-dose DSF groups.

3. As stated in the Introduction, DSF has garnered extensive attention on the Internet as a Lyme disease treatment. This raises the possibility that an "Internet placebo effect" might influence patient response to treatment (Ponten et al, Pain Reports 2019;4:e698), and no objective signs of clinical improvement such as change in QOL scales, seropositivity or immune markers are provided. The high level of adverse effects of DSF adds to the suspicion that results reported by the authors are based on a placebo effect. It would help to have a Table showing all the side effects of DSF treatment in these patients in order to compare these adverse events to patient symptoms. If symptoms and adverse events are similar, it is hard to judge whether there is a true benefit of this potentially toxic treatment.

4. The Results section states that 27 of the 67 patients were still on treatment, in contrast to the Discussion section statement on page 6 that "each patient was closely followed up through the completion of their treatments." This raises the issue of whether these 27 patients should be excluded from the study, since they apparently had ongoing symptoms of CLD. Again, the lack of objective response parameters makes the study "results" very difficult to interpret.

5. On page 8, the authors state that five patients achieved "enduring remission". How was this remission measured? Again the lack of objective parameters makes the outcome suspect. Another possible benefit is the effect of alcohol abstinence with DSF use. How many patients were drinking alcohol prior to the study? Although the authors suggest that DSF clearance may influence the response to treatment, no evidence for this mechanism is presented. In fact, a recent in vitro study could not demonstrate a benefit of DSF in treating stationary forms of B. burgdorferi (Alvarez-Manzo et al, Antibiotics 2020;9:542). Thus a placebo effect of this toxic treatment is a distinct possibility.

6. Tables: A large percentage of CLD patients take alternative treatments and supplements. These should be listed, and did they affect the study results? Was IRB approval obtained for the study, and if not, why not?

Minor Comments

1. Page 11: "Once a patient was deemed to be a good candidate for disulfiram treatment, the provider had an extended discussion with the patient and (if appropriate) his/her significant other(s) about..."

2. The authors should state whether patients in this study were included in previous reports of DSF treatment.

Author Response

Manuscript ID antibiotics-908951

Responses to Reviewer 3

1.The French study polled the ‘Lyme community’ seeking evidence of toxicity. The respondents were persons unknown to the authors who, ipso factohad not been either diagnosed or treated by them. The authors had no documentation of the provenance of the patients other than that obtained through the questionnaires and had no way of verifying the accuracy or truthfulness of the responses. The authors of the French study had no professional medical relationship with any of the respondents to their survey. The current mss. is a retrospective review of individuals who had been personally cared for by two of the co-authors, examined, an anamnesis personally obtained and, exhaustive laboratory testing undertaken using a wide range of available validated methods (e.g., methods ‘vetted’ and approved by the New York State Department of Health), reviewed all pertinent past records and followed them through their treatments as well as inquired as to their ongoing status beyond completion of therapy and if remaining on therapy, often beyond the the identifying period of March 2017 through March 2020 up until the original mss. was submitted, late July 2020. Although March 2020 was the limit for adding persons to be reported upon, that does not mean that those identified persons could not be followed further in order to inform the longer-term outcomes of the 67 patients identified during the3 years’ inclusionperiod that we used to identifypersons in whom disulfiram was applied. We believe there is a great deal of difference between the studies: our was a retrospective analysis of clinical care delivered directly by two of the co-authors of the present study over not only the 3 years analyzed, but often for months or years preceding application of disulfiram and up through the time of submission of our original mss. July 2020.We also find it perhaps telling that the French study found both toxicities as well as benefits but the title of the paper reflects only toxicities.

2.We agree with Reviewer 3 that manyof the patients in our study could be characterized as having chronic Lyme disease and some also had evidence of co-infection with either babesiosis or bartonellosis or both. These co-infections as well as additional pertinent medical conditions are indicated in Table 1. The paper by Shor et. al. more closely aligns with our concept of chronic Lymedisease than the Stricker & Fesler article which conflates chronic Lyme disease with other co-infections as opposed to the Shor et. al. paper which restrictedchronic Lyme disease as being due to Borrelia burgdorferi sensu lato,per se. Surely, chronic Lyme disease can co-exist with many other tick-or vector-borne diseases. We will incorporate mention of the Shor.definition of chronic Lyme disease which we feel does comport with mostof the 67 subjectsin the reportand will also reference the Stricker & Fesler definition as well as Rebman & Aucott’s PTLDS. We presented the co-infections and complicating co-morbid conditions in our original Table 1. A detailed review and recitation of each of the 67 cases, we feel would compriseadditional lengthy verbiage –reportage of the clinical

history of our Cases 1-3 in our original article required 10+ pages in the final published version. We hope the editor, publisher and reviewers can grant, based on our background and experience in this field, that the cases ‘qualified’ as bona fidecases of persons with Lyme disease with supportable histories and with reasonable laboratory corroboration. Although we could provide detailed documentation for each of the 67 cases, this will be a time consuming task and would or will require likely months-long delay in possible publication of the current mss.,the substance of which, we feel, is timely and ‘time-sensitive’. Any mss. can be improved –and in fact we are modifying the mss. based on comments of all reviewers -we believe a detailed precis’ of each case is superfluous and unnecessary.

3.We thank Reviewer 3 for bringing to our attention the Ponten et. al. study which demonstrates that placebo-effect can be induced even with internet-based-therapy (IBT). Reviewer 3 posits that the interest evoked in disulfiram as reflected by membership numbers of disulfiram-focused Facebook groups might be a placebo-related phenomenon. This is speculative for a number of reasons. We can not know which of the Facebook group members are being treated by healthcare practitioners as opposed to “Do It Yourself” (DIY) approaches with internet-secured disulfiram without any clinician’s prescription or supervision. If under the care of a health care practitioner, it is impossible to know if that is direct face-to-face care or, instead IBT. We acknowledge the importance of placebo-effect as well as, in our study, the possibility both that patients might have been influenced to exaggerate their degree of improvement in order to try to ‘please’ the clinicians as well as the possibility that the clinicians unconsciously exaggerated the perceived benefit of application of disulfiram in the patients in whom we have reported, and despite the best efforts of both clinicians and patients to remain ‘objective’. As this was not a formal research study, standardized symptom or status rating scales were not used but we agree fully with Reviewer 3 that such data, as might be readily incorporated in a formal research trial, would yield abundant data that would be less subject to recall bias. We believe that the disulfiram trial out of Columbia University headed by Dr. Brian Fallon incorporates such rating scales and when completed, will provide more ‘robust’ data. As clinicians interacting with our patients we rely heavily onwhat patients report to us, which we always endeavor to faithfully record in the record and this is evident in our voluminous hand written data. Regarding their status and their estimation of ‘net’ benefit of application of disulfiram or not, well, we simply asked the patients directly. Although many experienced adverse effects and we have been scrupulous to report these, nonetheless, the perceived net benefit is clear. Even Case 7, who developed very painful peripheral neuropathy some 9 weeks in to very high dose disulfiram (some 9.4mg/kg/day) fully recovered from the neuropathy and has remained clinically well for more than one year off all anti-infective treatment. Was it worth it to her? Well, wehave asked her that question directly and she answered resoundingly in the affirmative. Prior to disulfiram, whenever antimicrobial treatment was discontinued, her symptoms relapsed within days to weeks resulting in a demoralizing requirement to resume

both antibiotics as well as anti-parasitic therapy with azithromycin and atovaquone, the latter an extremely expensive agent in the United States. In virtually all cases (with a few exceptions of persistent mild neuropathic residua) toxic effects were fully reversible. We believe the ‘toxicity’ of disulfiram has been exaggerated in order to sow fear and to undermine itscompetitive advantage in the field of Lyme diseaseresearch to advancetherapeutics. This has implications for researchersas theystruggle to secure their share of limited private and/or government funding. Toxicities of drugs are a relative matter. To be borne in mind include the seriousness of the illness being treated as well as the toxicities of other available therapies. Chronic and neurologic Lyme disease is a very serious condition that can lead to death if untreated (Liegner KB, Duray P, Agricola M, Rosenkilde C, Yannuzzi L, Ziska M, Tilton R, Hulinska D, Hubbard J, Fallon B. Lyme Disease and the Clinical Spectrum of Antibiotic-Responsive Chronic Meningoencephalomyelitides. J Spirochetal and Tick-borne Dis 1997;4:61-73. See attached PDF). Intensive treatment is often necessary to prevent deterioration of the patient. Prior to the recognition of disulfiram’s possible utility, manyclinicians and academicianshave had to resort to intensive antimicrobial therapies which carry their own very well-documented suite of adverse and toxic reactions including agranulocytosis/ aplastic anemia, line-sepsis and line-phlebothrombosis, C. difficilecolitis, liver injury, gallbladder complications requiring surgery and many others, some of whichhave resulted in fatality. By comparison, in our small cohort of patients followed closely and with what we felt was ‘judicious’ dosing, virtually all adverse effects were reversible and in no case was there anycatastrophic complication nor fatal toxicities. We recognize the potential toxicity of disulfiram which has been well-reported in the worldwide peer-reviewed scientific literaturein relation to its use for the treatment of alcoholism. We felt obliged in our May 2019 publication to bring these to the attention of treating clinicians so they couldbe mindful of these and vigilant to detect complications early in order to intervene early (usually bysuspending disulfiram)to try to avert serious injury. We hopethat the conduct of our practice in employing disulfiram cautiously, with informing both of the patients an also of their other clinicians, with regular laboratory and clinical monitoring may serve to a degree as a ‘model’ for how to apply disulfiramresponsibly,attempting to maximize benefit and minimize risk.

4.We have modified the mss. so as to indicate that patients who were identified for analysis during the period March 2017 through March 2020 might also bequeried beyond March 2020 about their ongoing clinical status. We have clarified that the 3 year period identified that cohort ofpatients whose status and course were to be analyzed but not to ‘restrict’ our ability to follow their course further out and up through the time of submitting our final mss. for possible publication. Once again, this is not a formal research study and we do not feel we are ‘locked in’ to a rigid protocol as would be likely a requirement of an IRB-approved research study –again this is care provided during the ordinary course of our clinical practice of medicine.

5.We reiterate: we asked the patient as to their status and whether or not they experienced relapse of symptoms or had required reinstitution of antimicrobial

therapy whether from us or other practitioners. In our experience dealing with persons with chronic Lyme disease and associated co-infections, when symptomatic such patients would typically return for further care driven to do so by the unpleasant nature of their symptoms and personal suffering, seeking the relief that application of antimicrobial agents usually afforded. It was striking to us that our usual ‘repeat customers’ stoppedmaking their usual follow-up appointments in stark contrast to their years of prior care for their chronic relapsing symptoms. With our first three patients who each had very difficult and very well-documented illness they required no further care and Dr. Liegner even joked to his staff “We’re destroying our practice!” But of course, as physicians we want and hope our patients will get fully well to the point that they are not dependent upon us and can go on to live full, vibrant and healthy lives. That is the goal of the practice of medicine after all. It is just that with the heretofore standard antimicrobial approaches (even very intensive ones) this has not been within our power to achieve. We are not claiming that disulfiram is some kind of ‘miracle cure’. We do not feel it is the ‘panacea’ for Lyme and other co-infections. Merely we have found and continue to find that it is a potent agent and a newly discovered ‘tool’ for our ‘tool-box’, our armamentarium of treatment for Lyme disease and perhaps for babesiosis and even bartonellosis (more data needed for both of the latter infections). We also believe that other agents are coming down the pike which may be equal to or better than disulfiram and perhaps easier to apply and with fewer potential adverse effects. Finally, none of our patients had any problem with excess alcohol consumption. None were alcoholics and any benefit derived from disulfiram use can in no way be imputed to cessation of alcohol ingestion. The study by Alvarez-Manso et al used unique methods that were not comparable to those of Pothineni et. al. in their report in Drug Design, Devlopment and Therapy.and conclusions of the Alvarez-Manzo et.al. study are open to question. Also Potula et. al. recently published their in vitroand in vivoresults(Potula H-H SK, Shahryari J, Inayathullah M, Malkovskiy AV, Kim K-M, Rajadas J. Repurposing Disulfiram (Tetraethythiuram Disulfide) as a Potential Drug Candidate against Borrelia burgdorferi In Vitro and In Vivo. Antibiotics2020,9(9), 633;https://doi.org/10.3390/antibiotics9090633which serve to contradict and throw in to doubt the report of theAlvarez-Manzo et. al.group and this discrepancy needs to be reconciled. That disulfiram forms colloidal aggregates above a certain concentration might, plausibly explain the difference. In the fullness of time, the relative merits of the competing therapeutic approaches will become clearer.Regarding the issue of objective markers and in further clarification to issues raised in point 5, above, we did not routinely obtain serial serologies nordirect antigen detection testing (e.g. polymerase chain reaction [PCR] tests for detection of Lyme disease, babesiosis or bartonellosis or fluorescent-in-situhybridization testing [F.I.S.H.] for babesiosis or bartonellosis) because this would have entailed added expensefor patientsto bear.As many of the patients have enjoyed very improved quality of life after or while undergoing disulfiram treatment, they had little motivation to pursue serial testing. One exception is Case 3 of the original report who experienced symptoms of relapse

and who requested testing which, in fact, revealed a positive Lyme PCR which correlated with his clinical status. Surely, funded research project ought to incorporate serial testing including serologies, direct detection methods and any useful biomarkers or indirect indicators of ongoing infection of inflammation. As clinicians caring for ill persons, we did not feel it would be appropriate to insist on such testing if patients perceived no need for it in order to determine their care. The clinical trial under the auspices of Columbia University which we believe includes serial laboratory and formal validated patient self-assessment scales is presently ‘on hold’ due to COVID-19. I understand from Dr. Fallon that he is seeking a modification of his protocol to allow tele-medicine so that the trial can resume.

6.We were stringent in requiring patientsavoid supplements and herbals for which very little interactions data is available.We can only thinkof one patient who was determined to continue a few supplementsprovided by another practice throughout the course of disulfiram treatment. So, effectively, confounding effects of the multiplicity of supplements and herbals that characterize some practices which deal with tick-borne illness,wasa ‘non-issue’ in the patients we report.In the United Statesgenerallyand specifically within New York State, off-label use of previously Food and Drug Administration (FDA)-approved medications is within the prerogative of the treating physician and does not require any regulatory approval. Although it may be ‘novel’, our care of the individuals described in the current report it is not considered ‘experimental’ or ‘research’ per se. This was not a ‘study’. This was not a ‘research project’. This was not an Institutional Review Board (IRB)-approved research study nor was IRB-approval necessary or required. Although the authors of the 2019 study and the current study maintain hospital affiliations none is a hospital or a government employee and the conduct of their practice is independent of their affiliated hospitals. Publications which ensue on experiences derived from the ordinary course of clinical practice do not require ‘pre-review’ by the affiliated hospital or hospital systems. Were they truly ‘research’,IRB-approval would be required.. This was not the case in the work reported in MDPI-Antibiotic ID 908951. We never sought to obtain any signed informed consents for use of disulfiram from any of the patients treated in the 2019 publication nor in those in the currentreport. However, each patient was fully informed about potential risks, potential benefits as well as uncertainties about the use of disulfiram in the setting of Lyme and other tick-borne disease and this is amply reflected in the medical record of each patient. Additionally, prior to employing disulfiram, it was our practice to correspond in writing toeach of the otherpractitioners with significant involvement in the patient’s care. Each patient received a copy of the letter and this also served to document careful informing of the patient. This is mentioned in lines 244-253 of the current mss. We have subsequently learned that some practitioners have opted to require patients to sign a formal informed consent prior to use of disulfiram. We have not done so.

Minor

comments:

1, Page 11: it is unclear to the authors just what Reviewer 3 is seeking us to address. Perhaps the Reviewer can clarify this so that we can make an intelligent response.

2. We agree that it would be sensible and correct to clarify which patient in the current mss. correlate to Cases 1,2 and 3 inour original report. Enumeration is different but we do agree and we will endeavor to clarify which of the current mss. Case numbers represent Cases 1,2 & 3 from the May 2019 publication(for the information of the reviewer they correspond to Cases 18, 49 and 10 respectively)..

Round 2

Reviewer 1 Report

I am not satisfied with the rebuttal of the authors with their insistence that their observations of this study is as close to clinical without being a clinical study . This is my main concern among others and reject this manuscript .

Author Response

Reviewer 1 flatly rejects our mss.and does not entertain the value of reports of observational experiences of clinicians caring directly for patients nor acceptance of work that is done outside of an IRB-approved RCT. We reiterate our mss. represents an analysis of experiences obtained during the ordinary course of clinical care and NOT as part of a research study. No IRB-approval nor formal informed consent document was required from a legal, regulatory, ethical nor moral point of view. Nonetheless, patients were very well informed and this is thoroughly documented in their clinical records.

As we tried to indicate, with support for the peer-reviewed literature and an historical example, much value can attach to just such reports andRCTs, important as they are in many situations, are far from the only source of truth in clinical matters or in medical science. Serendipitous and anomalous observations have sometimes ‘up-ended’ comfortable but erroneous assumptions in Medicine generally and in the field of Lyme disease in particular.

We thank Reviewer 1 for their efforts and engagement in the review of our mss., however, we disagree with the decision and we regret that we were unable to persuade the reviewer to our point of view.

Kenneth Liegner

Reviewer 2 Report

Appreciate the changes made to the manuscript. All the best!

Author Response

Manuscript ID antibiotics-908951The co-authors thank Reviewer 2 for support of our mss.and the further support for our revised mss.which incorporated modifications in view of constructive criticisms of the Academic Editor and reviewers. We believe this hasimproved the mss.which we hope may be accepted and published by MDPI-Antibiotics in due cours

Reviewer 3 Report

When a manuscript requires a seven-page response to reviewer comments and has to be completely rewritten, you know there is a problem. The current version reads more like a gossip column from a news magazine, and the scientific shortcomings outlined in the previous review are still glaringly evident despite all the bells and whistles. The authors need to shorten the manuscript and simply present their interpretation of the findings, which still lack objective support. Lack of IRB approval for the study and lack of specific patient consent for DSF treatment may be an ethical problem as well.

Two additional points:

  1. The authors prefer the CLD definition of Shor et al. and include the PTLDS description of Aucott et al. However their patients do not fit either of these definitions because many had coinfections, and therefore fall outside the restrictive CLD definition of Shor et al, and previous antibiotic treatment was not mentioned, so not necessarily PTLDS. Thus only the CLD definition of Stricker and Fesler applies to all of their patients.
  2. Table 3 does provide a risk-benefit outline, but the risks are described in more detail than the benefits. Perhaps the benefits could be presented in more detail as well.

Author Response

Response to Reviewer 3

RE: Manuscript ID antibiotics-908951Rather than respond to the substantive issues which the authors brought forward considering Reviewer 3s original comments, Reviewer 3 has resorted to what are in essence ad hominemattacks.

That the authorsresponses required 6 pages reflected a full-throateddefense of the value of observational reports, with support from the peer-reviewed literature (Berger et. al. reference, Lilie et. al. reference and one historical example Beaumont and Alexis St. Martin) does not indicate any weakness of the mss. The reviewer fails to address this rebuttal. The revision of the mss. was undertaken in response to the Academic Editorssuggestions and what we perceived as constructive criticisms of the Reviewers, including Reviewer 3. We believe these revisions haveimproved the mss. It isoddthat Reviewer 3 chooses to interpret the mere revision of a mss. in reviewper se-which is a very common scenario-as anindication of inherent weakness of the mss.

There was no legal, regulatory, ethicalor moral imperative for thecare of the patientsdescribed in themss.tobe under the supervision ofan Institutional Review Board nor torequire asigned ‘informed consent document.Nonetheless, patients were thoroughly informed of potential risks, benefits and uncertainties in the use of disulfiram and this is well documented in the clinical record of each patient. This is no different than in the clinical care of patients for whom other FDA-approved agents are employed, including commonly used antibiotics.

Each of the patients included in the report, in our estimation, suffered from chronic Lyme disease: e.g. persisting symptoms of Lyme diseasein well characterized patientsenduring 6 months or more despite prior application of antibiotic treatment. We listed the two published definitions of chronic Lyme disease(Shor et. al. and Stricker & Fesler)as well as Rebman and Aucotts definition of PTLDS because these subsume the types of individuals such as those whom we evaluated and treated. Furthermore, Shors definition of chronic Lyme disease acknowledges that persons can have chronic Lyme disease while suffering concurrently from other tick-or vector-borne co-infections. Shors definition of chronic Lyme disease due to ongoing infection with Bb sensu lato. This condition may exist with or without co-infection and does correspond to the individuals we treated. Their definition relied upon citationsfrom the published peer-reviewedliterature in which persistent infection was proven byPCR or other direct detectionmethods (e.g. culture, immunohistchemistry).Thislevel of proof is not oftenpossible to achieve ina given patient in the clinical setting.

We thank Reviewer 3for their efforts and engagement in the review of our mss., however, we disagree with the Reviewersexpressed points of viewand we regret that we were unable to persuade the reviewer of the value of the work orof thelatest version of the msswhich, in our opinion, is satisfactoryas is.          

Thank you,

Kenneth Liegner
